# Prevalence of Plasmid-Associated Tetracycline Resistance Genes in Multidrug-Resistant *Escherichia coli* Strains Isolated from Environmental, Animal and Human Samples in Panama

**DOI:** 10.3390/antibiotics12020280

**Published:** 2023-01-31

**Authors:** I. E. Ramírez-Bayard, F. Mejía, J. R. Medina-Sánchez, H. Cornejo-Reyes, M. Castillo, J. Querol-Audi, A. O. Martínez-Torres

**Affiliations:** 1Experimental and Applied Microbiology Laboratory, Vice Rectory of Research and Postgraduate Affairs, Universidad de Panamá, Panama City 0820, Panama; 2Water Microbiology Laboratory, Vice Rectory of Research and Postgraduate Affairs, Universidad de Panamá, Panama City 0820, Panama; 3Master in Environmental Microbiology, Faculty of Natural and Exact Sciences and Technology, Universidad de Panamá, Panama City 0820, Panama; 4Panamá Canal Authority, Apdo 526725, Panama; 5Sistema Nacional de Investigación (SNI), SENACYT, Panama City 0816, Panama

**Keywords:** *E. coli*, multiple antibiotic resistance, tetracycline, *tetA*, *tetB*

## Abstract

Antimicrobial resistance bacteria are nowadays ubiquitous. Its presence has been reported in almost every type of source, from water for agricultural and recreative use, water distribution pipes, and wastewater, to food, fomites, and clinical samples. Enterobacteriaceae, especially *Escherichia coli*, are not the exception, showing an increased resistance to several antibiotics, causing a global health and economic burden. Therefore, the monitoring of fecal microbiota is important because it is present in numerous reservoirs where gene transfer between commensal and virulent bacteria can take place, representing a potential source of resistant *E. coli*. In this work, antibiotic resistance profiles of 150 *E. coli* isolates from environmental, animal, and human samples, collected in three rural areas in Panama, were analyzed. A total of 116 isolates were resistant to at least one of the nine antibiotics tested. Remarkably, almost 100% of these exhibited resistance to tetracycline. Plasmid-associated *tetA* and *tetB* genes were detected in 42.86% of the isolates analyzed, *tetA* being the most prevalent. These results suggest that tetracycline resistance would be used as a convenient indicator of genetic horizontal transfer within a community.

## 1. Introduction

Antimicrobial resistance poses a health threat to the population at large. Worldwide, we are observing higher rates of morbidity and mortality, causing a severe social and economic burden in terms of cost, treatments, and hospital stays [1]. In fact, we are currently facing the proliferation of multidrug-resistant bacteria, mainly caused by the overuse of antibiotics, which include human, veterinary, and agricultural practices [2,3,4]. Specific pathogenic bacterium, and the complex microbial communities that inhabit the skin and mucous membranes cause extraintestinal infections that can remain for long periods of time, thus, promoting the transfer of resistance elements to other members of the microbiota [5,6].

Moreover, inappropriate and prolonged use of antibiotics in public health, veterinary treatments, and a lack of proper waste handling from the pharmaceutical industry have also influenced the natural selective pressure in bacteria present in the environment, promoting the emergence of antibiotic resistance [7]. Apart from natural resistance and spontaneous mutations, the acquisition of these determinants of resistance also takes place via horizontal gene transfer through either transformation, conjugation, or transduction [8].

*E. coli*, which is often associated with human and animal infections, is not an exception, and resistance to ß-lactams, fluoroquinolones, aminoglycosides, tetracyclines, sulphonamides, phenicols, and polymyxins has been reported worldwide (reviewed in [9]). In particular, resistance to tetracyclines shortly appeared after their first clinical use in both Gram-positive and Gram-negative bacteria [10,11,12], and rapidly increased as a result of plasmid and transposon horizontal transfer [13]. Like most antibiotics, several mechanisms of resistance to tetracyclines have been characterized, including active efflux pumps, enzymatic modification of the ribosomal target, decreased drug permeability, mutations, and enzymatic degradation of the antibiotics, the first two being predominant mechanisms. Nine tetracycline efflux genes have been identified in *E. coli*: *tetA*, *tetB*, *tetC*, *tetD*, *tetE*, *tetG*, *tetJ*, *tetL*, and *tetY* [14]. However, the predominant genes encoding efflux pumps in this microorganism are *tetA*, *tetB*, and *tetC* and they often appear together [9]. 

Although there are official reports that analyze the presence of multidrug-resistant *E. coli* in Panama, these studies are limited to clinical settings, and information regarding environmental samples is very scarce [15]. For this reason, in this study we addressed the analysis of multi-resistant *E. coli* strains isolated from water, and human, swine, chicken, and cow feces in three rural areas of Panama, as well as the prevalence of plasmid-associated tetracycline resistance genes, *tetA* and *tetB* in these isolates.

## 2. Results

### 2.1. High Percentage of the E. coli Isolates Exhibit Antibiotic Resistance

Sensitivity testing to nine antibiotics was performed on 150 *E. coli* isolates from fecal samples (chicken, cow, swine, and human) and water sources from the three different locations.

Resistance to at least one of the antibiotics analyzed was observed in all the sampling sites, representing 77.33% of the isolates (116 of 150) (Table 1). The highest percentage (90%, 45 of 50) of samples containing antibiotic-resistant isolates was observed in Ciudad del Niño, followed by Escobal (76%, 38 of 50) and, finally, El Arado (66%, 33 of 50). Regarding the sampling source, all the isolates from pig manure were resistant to one or more antibiotics, followed by isolates from chicken manure (90%, 27 of 30), cow manure (83.33%, 25 of 30), water (60%, 18 of 30) and human feces (53.33%, 16 of 30).

Ciudad del Niño isolates presented the highest number of antibiotic resistance (eight/nine antibiotics), followed by El Arado (five antibiotics) and Escobal (three antibiotics). Isolates from Ciudad del Niño presented the highest tetracycline resistance prevalence (88%, 44 out of 50 strains) compared to El Arado (76%) and Escobal (64%) isolates (Figure 1).

Resistance to gentamicin and amikacin (2%) was only present in Ciudad del Niño (Figure 1a). Levofloxacin and ciprofloxacin prevalence was the highest in Escobal (46% each; Figure 1b). In El Arado, only tetracycline (76%), trimetropin- sulfamethoxazole (8%), and chloramphenicol (8%) resistant strains were detected (Figure 1c).

Among all the isolates that exhibited resistance to at least one antibiotic, 98.28% of them were resistant to tetracycline, followed by trimetropin- sulfamethoxazole (53.45%), ciprofloxacin (29.31%) and levofloxacin (25%). Resistance to imipenem was not detected in any of the *E. coli* isolates analyzed (Figure 1d).

### 2.2. Multidrug Resistance Profiles

Multidrug-resistant *E. coli* isolates (resistance to ≥ three antibiotic classes) were identified in 41.38 % (48 of 116) of the samples. A total of 12 resistance unique patterns were observed (Figure 2). Tetracycline resistance was present in all the multidrug resistance patterns, followed by resistance to SXT (11 patterns) and CIP (10 patterns). 

The most frequent multidrug resistance patterns were pattern 5 (18 strains, CIP, LVX, SXT, and TET resistance), pattern 2 (13 strains, CHL, SXT, and TET resistance), and pattern 9 (7 strains, CAZ, CIP, LVX, SXT, and TET resistance). The remaining patterns were found in 1 or 2 strains.

Multidrug-resistant strains were detected only in two sites: Ciudad del Niño (25 of 48 multidrug-resistant strains) and El Arado (23 of 48). Isolates from chicken manure (18 of 48), pig manure (15 of 48), cow manure (7 of 48), water (7 of 48), and human feces (1 of 48) presented at least one multidrug resistance pattern (Appendix A).

### 2.3. Tetracycline Resistance

TET-resistant strains represented 76% of the *E. coli* isolates (114 of 150) (Figure 3). 

All the isolates collected from pig manure were resistant to TET (30 strains), followed by chicken (27 strains), cow (24 strains), water (18 strains), and human feces (15 strains). Tetracycline-resistant *E. coli* strains were found in the three communities. The highest number was found in Ciudad del Niño (38.60%, 44 of 114), followed by Escobal (33.33%, 38 of 114) and El Arado (28.07%, 32 of 114).

### 2.4. tetA and tetB Are Present in Nearly 45% of the Samples Analyzed

Since genes conferring resistance to tetracycline are mainly contained in plasmid DNA, the presence of plasmid-associated *tetA* and/or *tetB* was analyzed by multiplex PCR (Figure 4). 

A total of 85 *E. coli* isolates carrying plasmid DNA were detected by agarose gel electrophoresis after plasmid extraction and purification using chemical methods(see Appendix A). Out of them, 46 contained either *tetA*, *tetB,* or both genes (Table 2).

A total of 58.70% (27 of 46) were only positive for *tetA*, a 6.52% for *tetB* (3 of 46) and both genes were present in 34.78% (16 of 46) of the isolates analyzed (Table 2), thus *tetA* being the most prevalent.

The highest number of isolates containing *tetA* and/or *tetB* genes was detected in Ciudad del Niño (39.13%, 18 of 46), followed by El Arado (34.78%, 16 of 46) and Escobal (26.09%, 12 of 46).

When analyzing the resistance gene distribution per collection source, positive samples either from chicken or swine represented 23.9% (11 of 46), 21.7% from cows (10 of 46), 17.4% from water (8 of 46), and 13% from human feces (6 of 46 each) (Table 2).

The *tetA* gene was detected alone or in combination with *tetB* in 21.7% of the chicken, swine, and cow samples (10 of 46), 17.4% in water samples (8 of 46), and 10.9% isolates from humans (5 of 46). The *tetB* gene showed the highest prevalence in swine with 10.9% of positive samples (5 of 46), followed by isolates from chicken and cow with 8.7% (4 of 46) and 4.1% from human and water samples (3 of 73) (Table 2).

## 3. Discussion

Environmental contamination with bacterial pathogens, often through feces, poses a health risk, especially in developing countries, where water quality, sanitation, and hygiene needs to be improved [16]. Panama faces this situation, especially in rural areas where humans and animals live near each other, which possibly facilitates the transmission between them, furthering the absence of household wastewater sanitation systems. Thus, monitoring of pathogenic species with antibiotic resistance factors in the environment is essential to prevent the local and, even, global spread of such resistant bacteria. 

Recent studies correlate high levels of antibiotic consumption with high levels of resistance worldwide, especially in low and middle-income countries [17]. In general, we have found similar results on a local scale: Ciudad del Niño, where veterinary care is continuous, as is the control of feeding and maintenance of animals in galleys, presented the greatest multidrug resistance. On the other hand, the strains isolated in El Arado and Escobal (where animals are fed mainly for their owners’ consumption, and thus exposure to antibiotics is very low) were the ones with the lowest antibiotic resistance rates. On the other hand, differences between antibiotic-resistant isolates found in swine and cows (cattle) from one location to another are not so significant, despite the different veterinary treatments. These results do not agree with those published in other geographical areas, such as Europe, where a clear difference is observed in the resistance patterns of *E. coli* isolated from captive-bred animals and organic ones [18].

Although differences are observed in terms of impact in each area, due to the different conditions described above, the resistance pattern observed is quite similar despite the source. Globally, the *E. coli* isolates presented greater resistance to tetracycline, trimethopin/ sulfamethoxazole, followed by chloramphenicol and amikacin, then by ceftazidime, levofloxacin and ciprofloxacin, and finally to gentamicin and imipenem. Similar results have been reported in various studies [19,20]. Multidrug resistance patterns related to livestock, poultry, and water sources have been detected in *E. coli* including tetracycline + trimethopin/sulfamethoxazole + ciprofloxacin, tetracycline + trimethopin/sulfamethoxazole + ciprofloxacin + gentamicin, ciprofloxacin + trimethopin/sulfamethoxazole + chloramphenicol and tetracycline + ciprofloxacin + chloramphenicol like those found during this analysis. Other similar patterns additionally include the combinations of levofloxacin, ampicillin, cefotaxime, nalidixic acid, nitrofurans, cephalosporin, and colistin, some of these antibiotics were not used in this study [21,22,23].

Runoff and livestock wastewater discharge are potential pathways for the dissemination of antibiotic-resistance genes in soil and water environments [24]. Regarding the results obtained from the isolation of water sources, we presume that the differences observed in resistance are due to climate and animal waste management in each sampling site. For example, in Escobal, where the incidence of resistant strains was lower, samples were collected during the dry season thus the lack of runoff could have reduced the contribution of resistant strains from contaminated soil to the already decreased water flow of near creeks and streams. Whereas, in Ciudad del Niño, samples were collected in treatment ponds, where animal waste accumulates. This could explain the higher number of resistant strains detected in this area. As Rosenblatt-Farrell pointed out, tailing ponds provide an alternative pathway by which birds and insects can pick up multidrug-resistant bacteria, spreading them into the environment [25].

In livestock waste, the classes of antibiotic resistance reported in the literature include tetracyclines, sulfonamides, macrolide-lincosamid-streptrogramin B, FCA (floroquinolone, quinolone, florfenicol, chloramphenicol, and amphenicol) and β-lactams. Tetracycline and sulfonamide resistance genes are generally the most abundant. Furthermore, resistance abundance is usually higher in swine and chicken waste than those in cows [24]. In this study, both tetracycline and sulfonamide (sulfamethoxazole) resistances were the most abundant in every sampling site. Moreover, swine and chicken samples presented a higher abundance of resistance as previously described.

As mentioned before, TET-resistant strains represented 76% of the *E. coli* isolates analyzed in this study, and 45% of them contained plasmid-associated *tetA* and/or *tetB* genes. In recent studies, an increase in the frequency of the *tetA* resistance gene has been observed, followed by the *tetB* gene, and to a lesser extent *tetC*, *tetD* and *tetE*, in farm animals, [26,27].

The acquisition of plasmid-mediated tetracycline-resistant *E. coli* could be given through contaminated food consumption from breeding animals or by the proximity of water sources to farms where animals are reared [28,29,30]. The prevalence of plasmids conferring tetracycline resistance in the cow samples in the three communities could be due to the intensive breeding system and the strict veterinary controls to care and fatten the farm animals, which makes them most vulnerable to acquire these mobile elements through their diet or through disease control [26,31]. Even though these genes are usually detected alone, they can also appear combined, which favors an increased mechanism of resistance and pathogenicity within the organism [30,32,33]. The little sanitization and proximity to the water source within these communities favor the high prevalence of resistance genes.

The high presence of antimicrobial resistance in the samples, linked to a high prevalence of tetracycline resistance genes, may indicate that the environment where the samples were taken is contaminated with antibiotics [34] and that there is a distribution of genes that encode for its resistance in *E. coli* [30,35]. The high prevalence observed can occur because these communities are dedicated to self-managed agricultural production [31]. These results suggest that tetracycline resistance might be a good indicator of genetic horizontal transfer within a community.

## 4. Materials and Methods

### 4.1. Sample Collection

A total of 150 samples from cow, swine, chicken and human feces, and water nearby (ten samples per source and area) were collected in three different rural areas of the country: (1) in the village of El Arado, district of La Chorrera, Province of Western Panama; (2) in Escobal, district of Colon, Province of Colon, and (3) in Ciudad del Niño, district of La Chorrera, Province of Western Panama.

Ciudad del Niño is an orphanage located in the province of Panama Oeste (Western Panama). It has poultry and swine farms, as well as a small cattle farm, all within a close perimeter of less than 1 km. Water samples were taken from the oxidation ponds of the swine farm, while the human samples were taken from the children who live in the orphanage. The sample collection points in El Arado and Escobal were more distant from each other. Chicken and swine samples came from domesticated animals of local people and cow samples were collected from local cattle farms. Water samples came from streams found in the area, and human samples were provided by the healthcare facilities in the three areas (Cirilo Escobar Health Center in Escobal, Centro de Salud in El Arado, and the orphanage health facility in Ciudad del Niño).

Stool samples were collected using sterile swabs and kept in peptone water. Water samples (100 mL) were collected in sterile plastic bottles. Samples were transported the same day to the laboratory and processed within the first 24 h.

### 4.2. E. coli Isolation and Confirmation

Isolation and confirmation protocols were adapted from the book “Microbiological Examination of Water and Wastewater” [36]. Briefly, the stool samples were inoculated in a Presence-Absence medium (10 mL, Difco, Tucker, GA, USA) using the sterile swabs previously kept in peptone water. Water samples (1 mL) were inoculated also in 10mL of Presence-Absence medium. Tubes were incubated at 37 °C for 24 h. Positive samples were streaked on Eosin-Methylene Blue agar (Acumedia, San Bernardino, CA, USA) and incubated at 37 °C for 24 h. Subsequently, a representative colony from each plate, with the characteristic metallic green sheen, was inoculated in EC MUG broth (Merck, Rahway, NJ, USA) and incubated at 37 °C for 24 h. The ones emitting UV fluorescence were selected and biochemical confirmation was carried out using the API 20E biochemical identification system (bioMérieux, Marcy-l’Étoile, France) according to the manufacturer’s instructions.

### 4.3. Antibiotic Susceptibility Testing

In order to determine the antibiotic profile of the 150 isolates, a Kirby-Bauer disk diffusion assay was employed using Müller Hinton agar plates according to the Clinical and Laboratory Standards Institute (CLSI) guidelines [12], using the following antibiotics: amikacin (AMK; 30 μg), ceftazidime (CAZ; 30 μg), chloramphenicol (CHL; 30 μg), ciprofloxacin (CIP; 5 μg), gentamicin (GEN, 120 μg), imipenem (IPM; 10 μg), levofloxacin (LVX; 5 μg), trimetropin/sulfamethoxazole (SXT; 25 μg) and tetracycline (TET; 30 μg).

Briefly, a bacterial inoculum was prepared from a 24 h culture on Trypticase-Soy agar (Bioxon, México), from which a portion of the culture was taken by swabbing. Then, swabs were resuspended in the saline solution until a turbidity of 0.5 on the McFarland scale (1.5 × 10^8^ CFU ml^−1^) was reached. A sterile swab was then impregnated with the adjusted bacterial suspension and spread in each Müller Hinton plate. Discs containing each antibiotic were applied onto the agar (five discs per plate) and plates were incubated for 24 h at 37 °C. Inhibition halos were measured and interpreted according to the categories established by the reference methods. The reference *E. coli* strains (ATCC 25922 and ATCC 35218) were included in each susceptibility determination. Inhibition zone breakpoints were interpreted according to information supplied by the manufacturer and the CLSI guidelines [37].

### 4.4. Plasmid DNA Extraction

Samples were processed using the alkaline lysis extraction method [38]. Briefly, after incubation of 18 h at 37 °C in Nutrient broth, cells were collected by centrifugation at 13,000× *g* for 5 min and resuspended in 100 μL of 25 mM Tris pH 8, 50 mM glucose, and 10 mM EDTA. After incubation for 5 min at room temperature (RT), 200 μL of 0.2 M NaOH and 1% *w*/*v* SDS were added, and samples were placed on ice for 5 min. Then, 50 μL of cold glacial acetic acid was added, and samples were incubated on ice for another 5 min. After centrifugation for 10 min at 13,000× *g*, the supernatant (approximately 450 μL) was transferred to a clean microtube and 400 μL of chloroform was added. The solution was mixed by inverting the tube followed by centrifugation at 13,000× *g* for 5 min. The top layer was then transferred to a clean microtube, and two volumes of 100% cold isopropanol were added. The solution was centrifuged at 13,000× *g* for 10 min and the resulting pellet was washed with 500 μL of 70% ethanol and left at RT until dry. Finally, the pellet was resuspended with 50 μL of sterile nuclease-free water and treated with RNase for 30 min at 37 °C. The presence of plasmids was then determined by 1% (*w*/*v*) agarose gel electrophoresis, ethidium bromide staining, and visualization.

### 4.5. Detection of tetA and tetB Genes by Multiplex PCR

The following set of primers was used to amplify the *tetA* gene: 5′-GCTACATCCTGCTTGCCTTC-3′ and 5′-CATAGATCGCCGTGAAGAGG-3′ and *tetB*: 5′-TTGGTTAGGGGCAAGTTTTG-3′ and 5′-GTAATGGGCCAATAACACCG-3′, according to Bailey et al. (2010) [39]. PCR mix contained 2 μL of the sample, dNTPs (200 µM), primers (1 µM each), and ADN polymerase (1.25 U; Invitrogen, Walthamm, MA, USA) in a final volume of 25 μL. The following conditions for DNA amplification were used: 94 °C for 4 min, 94 °C for 1 min, 55 °C for 1 min and 72 °C for 1 min (35 cycles) and a final extension cycle for 5 min at 72 °C. Amplification products were visualized by ethidium bromide staining after 1.5% (*w*/*v*) agarose gel electrophoresis in 0.5X TBE buffer.

## 5. Conclusions

In this study, we have analyzed the presence of multidrug-resistant *E. coli* strains isolated in the environment in three rural areas of Panama. The results show the circulation of multi-resistant strains in all the analyzed areas as well as a high prevalence of tetracycline resistance genes in humans, animals, and circulating waters. Approximately half of the tetracycline resistance genes are associated with plasmids, which could favor their dissemination by horizontal transfer. Subsequent studies for the analysis of these plasmids and their conjugative nature should be carried out to corroborate their transfer mechanisms.

## Figures and Tables

**Figure 1 antibiotics-12-00280-f001:**
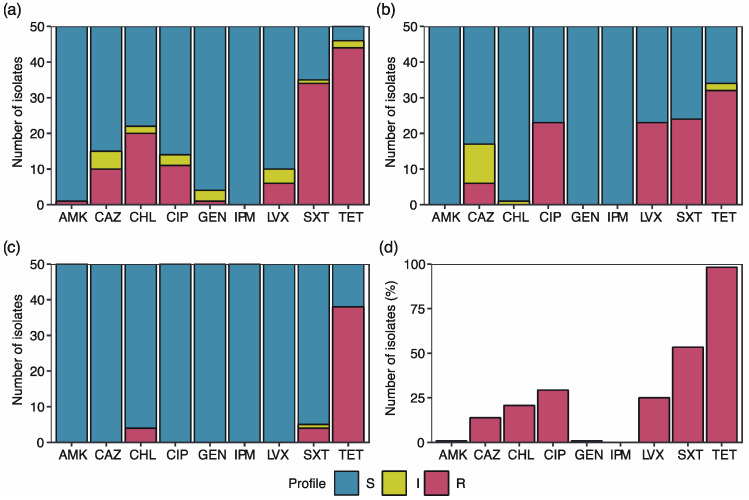
*E. coli* antibiotic resistance profile in (**a**) Ciudad del Niño, (**b**) Escobal, (**c**) El Arado: S indicates sensitivity to the antibiotic (in blue), I indicates intermediate sensitivity (in yellow) and R indicates resistance (in red). (**d**) Shows the whole percentage of resistant strains to each of the antibiotics analyzed. AMK: amikacin; CAZ: ceftazidime; CHL: chloramphenicol; CIP: ciprofloxacin; GEN: gentamicin; IPM: imipenem; LVX: levofloxacin; SXT: trimetropin/sulfamethoxazole; TET: tetracycline.

**Figure 2 antibiotics-12-00280-f002:**
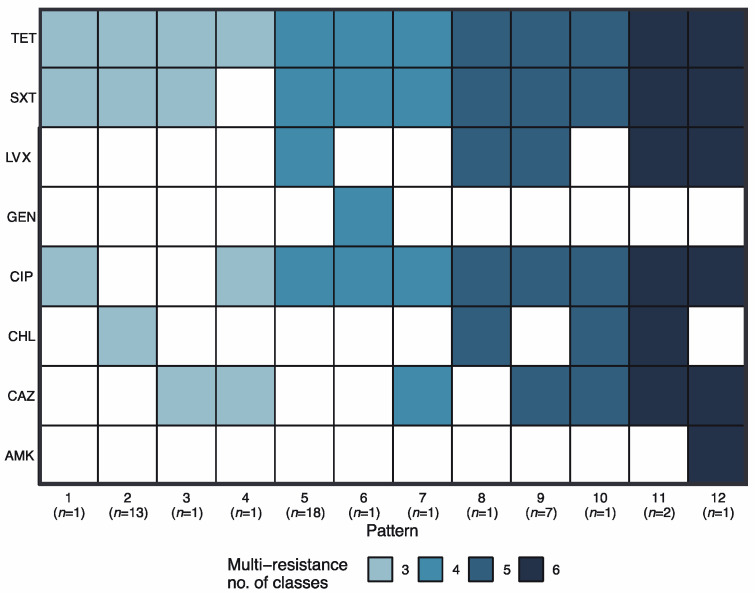
Multidrug resistance patterns detected in *E. coli* isolates. The colored boxes indicate resistance to each antibiotic (ordinate) creating a unique pattern (abscissa). The color gradient represents the number of resistances and n indicates the number of isolates belonging to each pattern.

**Figure 3 antibiotics-12-00280-f003:**
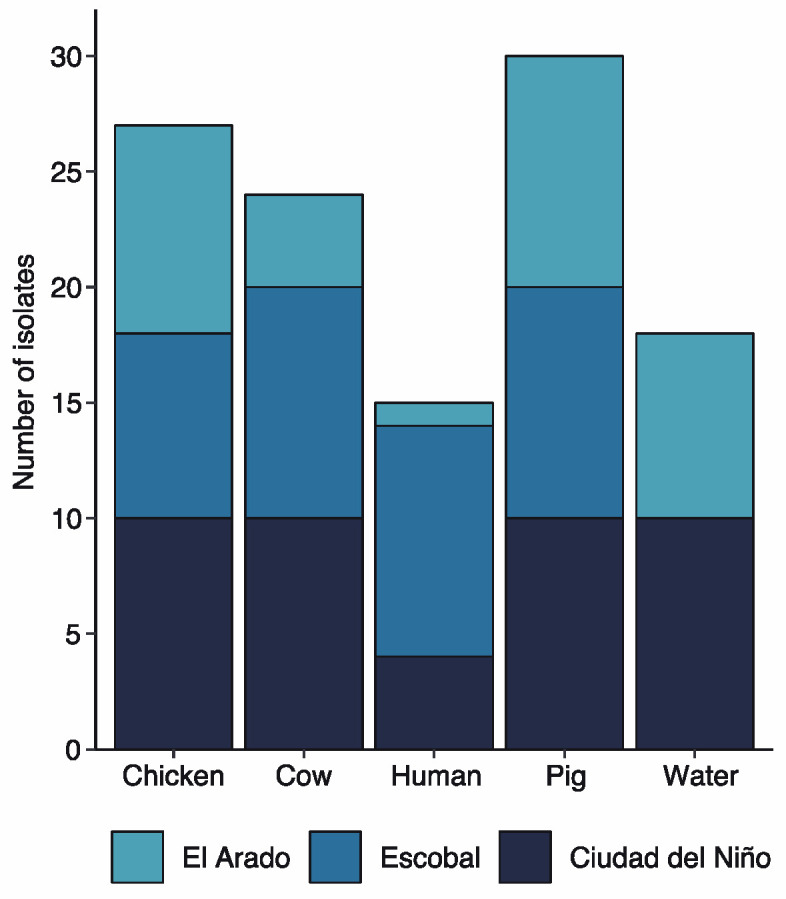
Tetracycline resistant strains distribution per sample source.

**Figure 4 antibiotics-12-00280-f004:**
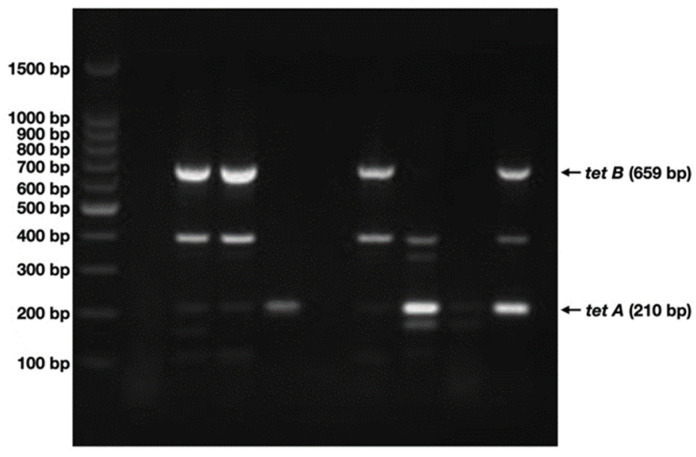
Detection of *tetA* and *tetB* by multiplex PCR. Agarose gel electrophoresis of representative negative, *tetA* and/or *tetB* positive samples is shown. Amplification fragments for each gene are indicated on the right and molecular weight markers on the left.

**Table 1 antibiotics-12-00280-t001:** Total antibiotic-resistant strains by source and sampling site.

Source	Site (%)			Total Antibiotic Resistance Strains (%)
	Ciudad del Niño	El Arado	Escobal	
Chicken	10 (6.67)	9 (6.00)	8 (5.33)	27 (18.00)
Cow	10 (6.67)	5 (3.33)	10 (6.67)	25 (16.67)
Human	5 (3.33)	1 (0.67)	10 (6.67)	16 (10.67)
Swine	10 (6.67)	10 (6.67)	10 (6.67)	30 (20.00)
Water	10 (6.67)	8 (5.33)	0 (0)	18 (12.00)
Total (%)	45 (30.00)	33 (22.00)	38 (25.33)	116 (77.33)

**Table 2 antibiotics-12-00280-t002:** Detection of *tetA* and/or *tetB* in the samples analyzed in this study. The number of positive isolates and the corresponding percentage (in parenthesis) are indicated per sample source and sampling area.

	*tetA* (%)	*tetB* (%)	*tetA* and *tetB* (%)	Total (%)
Chicken	7 (15.22)	1 (2.17)	3 (6.52)	11 (23.91)
Cow	6 (13.04)	0 (0)	4 (8.70)	10 (21.74)
Human	3 (6.52)	1 (2.17)	2 (4.35)	6 (13.04)
Swine	6 (13.04)	1 (2.17)	4 (8.70)	11 (23.91)
Water	5 (10.87)	0 (0)	3 (6.52)	8 (17.39)
Ciudad del Niño	9 (19.57)	3 (6.52)	6 (13.04)	18 (39.13)
El Arado	10 (21.74)	0 (0)	6 (13.04)	16 (34.78)
Escobal	8 (17.39)	0 (0)	4 (8.70)	12 (26.09)
Positive (%)	27 (58.70)	3 (6.52)	16 (34.78)	46 (100)

## Data Availability

Not applicable.

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
