# Peer review of "Prevalence of Plasmid-Associated Tetracycline Resistance Genes in Multidrug-Resistant Escherichia coli Strains Isolated from Environmental, Animal and Human Samples in Panama"

_antibiotics, 2023, doi:10.3390/antibiotics12020280_

Round 1
Reviewer 1 Report
Following are my observations:
(1.) Title: the title is ok.
(2.) Introduction: In introduction the authors did not explore the scientific literature on plasmid associated tetracycline resistance as per the title of the article. Title reflection must be observed in introduction and discussion. I recommend the authors please write information about tetracycline resistance and horizontal gene transfer. And also add references with updated information on tet A, tet B resistance. Furthermore, the authors have mentioned results in introduction kindly, delete it.
(3.) Materials & Methods: sample collection and methods are fine however, I recommend authors to add / write new subsection "Ethics statement". In this subsection must be written number and date of the protocol with which the Local Ethics Committee has allowed conducting this research.
(4.) Results need a major revision and molecular detection and distribution of tet A and Tet B genes from the Ciudad del Niño , Escobal, El Arado has not been elaborated in results. Please include the gel pictures. (PCR results can’t be submitted in supplementary material )
(5.) Discussion: its poorly written and needs a major modifications :-
1. Discuss your MDR findings with others
2. Explain your findings of resistant tetracycline genes and what has been done so far .
(6 ). References:
Some references are very outdated (e.g.; ref no 16) And should be replaced with update one.
Author Response
Following are my observations:
(1.) Title: the title is ok.
(2.) Introduction: In introduction the authors did not explore the scientific literature on plasmid associated tetracycline resistance as per the title of the article. Title reflection must be observed in introduction and discussion. I recommend the authors please write information about tetracycline resistance and horizontal gene transfer. And also add references with updated information on tet A, tet B resistance. Furthermore, the authors have mentioned results in introduction kindly, delete it.
Updated information regarding horizontal gene transfer and tet A, tet B resistance has been included in the Introduction.
(3.) Materials & Methods: sample collection and methods are fine however, I recommend authors to add / write new subsection "Ethics statement". In this subsection must be written number and date of the protocol with which the Local Ethics Committee has allowed conducting this research.
This study was exempt from animal ethics approvals since the farmers managed the animals; thus, ethics approval was not required. Unfortunately, at the time human samples were obtained, which were kindly donated by the health care facilities in the three areas, the National Ethics Regulation was not still approved (Law number 84, May the 14th, 2019). Therefore, we did not collect the samples neither any personal information from the donors.
(4.) Results need a major revision and molecular detection and distribution of tet A and Tet B genes from the Ciudad del Niño , Escobal, El Arado has not been elaborated in results. Please include the gel pictures. (PCR results can’t be submitted in supplementary material )
PCR results have been included in the main text. Also results regarding tet A, tet B resistance have been elaborated.
(5.) Discussion: its poorly written and needs a major modifications :-
- Discuss your MDR findings with others
- Explain your findings of resistant tetracycline genes and what has been done so far .
Discussion about both MDR and tetracycline resistance has been expanded including recent findings in other geographical areas.
(6 ). References:
Some references are very outdated (e.g.; ref no 16) And should be replaced with update one.
References have been outdated, as the reviewer requested.
Reviewer 2 Report
Ramírez-Bayard et al. described the antibiotic resistance profiles of 150 E. coli isolates. These isolates were environmental, animal, and human samples collected in three rural areas in Panama. 116 isolates were found resistant to at least one of the nine antibiotics tested. 114 of 150 were found resistant to tetracycline. The manuscript is overall in its good shape.
Major comments:
1. Change to color schemes in the main figures.
2. Introduction regarding Tc resistance genes needs to be furthered. A good paper discussing the Tc resistome needs to be incorporated (10.1007/s00018-009-0172-6). In addition, I would like the authors to summarize the identification and evolution of plasmid-borne Tc resistance genes in the field as well as under laboratory settings (an example is 10.1093/gbe/evz197).
3. Following up on plasmid-borne Tc genes, the authors mentioned that "A total of 85 E. coli isolates carrying plasmidic DNA were identified (data now shown)". I would like the authors to include these data in this manuscript. The authors also indicated that mobile elements were responsible for the spread of the Tc resistance genes. Are these plasmids conjugative?
Minor comments:
1. Lines 30–31, "important" → "severe".
2. Line 52, remove "is".
Author Response
Ramírez-Bayard et al. described the antibiotic resistance profiles of 150 E. coli isolates. These isolates were environmental, animal, and human samples collected in three rural areas in Panama. 116 isolates were found resistant to at least one of the nine antibiotics tested. 114 of 150 were found resistant to tetracycline. The manuscript is overall in its good shape.
Major comments:
- Change to color schemes in the main figures.
Figures have been changed to color scheme as the reviewer requested.
- Introduction regarding Tc resistance genes needs to be furthered. A good paper discussing the Tc resistome needs to be incorporated (10.1007/s00018-009-0172-6). In addition, I would like the authors to summarize the identification and evolution of plasmid-borne Tc resistance genes in the field as well as under laboratory settings (an example is 10.1093/gbe/evz197).
Updated information regarding tet A, tet B resistance has been included in the Introduction.
- Following up on plasmid-borne Tc genes, the authors mentioned that "A total of 85 E. coli isolates carrying plasmidic DNA were identified (data now shown)". I would like the authors to include these data in this manuscript. The authors also indicated that mobile elements were responsible for the spread of the Tc resistance genes. Are these plasmids conjugative?
Approximately half of the tetracycline resistance genes were associated with plasmids, so we suggest that this could favor their dissemination by horizontal transfer. However, the reviewer is right, subsequent studies for the analysis of these plasmids and their conjugative nature should be carried out to corroborate their transfer mechanisms.
Minor comments:
- Lines 30–31, "important" → "severe".
This has been changed
- Line 52, remove "is".
This has been removed
Reviewer 3 Report
Comments for authors
The manuscript entitled “Prevalence of Plasmid-Associated Tetracycline Resistance Genes in Multi-resistant Escherichia coli Strains Isolated from Environmental, Animal and Human Samples in Panama” described the isolation of E. coli from feces and environmental samples in three different locations in Panama. The study of antibiotic resistance was also included. Tetracycline resistance was also deeply investigated. The scientific soundness and contents of this manuscript are fine. However, the current version of this presentation can be improved. Some points of content should be adopted to avoid confusion and repeated mention.
Authors should check the writing style and consistency of “antibiotic-resistant E. coli”, “antibiotic resistance in E. coli”, “antibiotic resistance profiles”, “multidrug-resistant E. coli”, “multidrug resistance patterns”, “multiple antibiotic resistance”, etc. with uniformity.
Title
1. Line 3; change to “Multidrug-Resistant Escherichia coli”.
2. Capital letters for “Associated”, “Resistant”, and “Genes” as per journal format.
Introduction
1. Line 32: change to “multidrug-resistant bacteria” and kindly check the whole manuscript.
2. Line 56: Remarkably.
Results
1. Table 1 need more column and row to summarize the total resistance number and % resistance of AMR in each source and sampling site.
See the examples in
Table 1: https://doi.org/10.1016/j.scitotenv.2012.10.106
Table 3: https://doi.org/10.3389/fped.2021.670470
2. Line 93: remove “multi-drug”
3. Line 94: add a hyphen “antibiotic-resistant E. coli” and kindly check the whole manuscript.
4. Figure 1: the full names of antibiotics are required in the footnote. For example, AMK: amikacin; CIP: ciprofloxacin; etc.
5. Line 123/125: change to “plasmid DNA”. Kindly check the whole manuscript.
6. Line 129-142: the % positive and total positive number for tetA, tetB, and both genes should be included in Table 2 instead of text and left the significant messages.
Discussion
1. Line 153: “multidrug-resistant”
2. Line 154: “location”
3. Line 158: “multidrug resistance” or multiple antibiotic resistance”
Materials and Methods
1. Subsection no. must be added for each subsection title.
2. How long the samples were taken to the laboratory and how long the samples were kept before processing? Information should be added to the manuscript.
3. Kindly check with the human ethic committee of your institute whether the samples taken from human sources, especially from children in this study need the ethics statement and consent form or not. It is different from the stool collection in animal sources. It is very important!!!!
4. Kindly provide the data of children (age range, total number, statement from their parents to allow them as volunteer in this study, and necessary information).
5. Line 247: References for E. coli isolation and confirmation must be added. Is this method under the standard protocol of E. coli isolation and confirmation? If it is not, please explain!
6. Line: 275 “Plasmid DNA extraction”.
7. Line 277: kindly provided the name of “nutritive broth”. Is it Nutrient broth?
Conclusion: I have never seen this section in this current version. No text or message is available to summarize the overall studies in the final paragraph of the Discussion section. This section should be separated and included to summarize the overall studies and outcomes of this work.
Author Response
Comments for authors
The manuscript entitled “Prevalence of Plasmid-Associated Tetracycline Resistance Genes in Multi-resistant Escherichia coli Strains Isolated from Environmental, Animal and Human Samples in Panama” described the isolation of E. coli from feces and environmental samples in three different locations in Panama. The study of antibiotic resistance was also included. Tetracycline resistance was also deeply investigated. The scientific soundness and contents of this manuscript are fine. However, the current version of this presentation can be improved. Some points of content should be adopted to avoid confusion and repeated mention.
Authors should check the writing style and consistency of “antibiotic-resistant E. coli”, “antibiotic resistance in E. coli”, “antibiotic resistance profiles”, “multidrug-resistant E. coli”, “multidrug resistance patterns”, “multiple antibiotic resistance”, etc. with uniformity.
We have modified the writing style in the whole text.
Title
- Line 3; change to “Multidrug-Resistant Escherichia coli”.
This gas been changed
- Capital letters for “Associated”, “Resistant”, and “Genes” as per journal format.
This has been changed
Introduction
- Line 32: change to “multidrug-resistant bacteria” and kindly check the whole manuscript.
The whole text has been checked and modified accordingly
- Line 56: Remarkably.
This has been changed
Results
- Table 1 need more column and row to summarize the total resistance number and % resistance of AMR in each source and sampling site.
Percentages have been included in table 1 as reviewer requested
See the examples in
Table 1: https://doi.org/10.1016/j.scitotenv.2012.10.106
Table 3: https://doi.org/10.3389/fped.2021.670470
- Line 93: remove “multi-drug”
This has been removed
- Line 94: add a hyphen “antibiotic-resistant coli” and kindly check the whole manuscript.
An hyphen has been included and we have checked the whole manuscript for style consistency
- Figure 1: the full names of antibiotics are required in the footnote. For example, AMK: amikacin; CIP: ciprofloxacin; etc.
As the reviewer requested, the full names have been included in the figure legend.
- Line 123/125: change to “plasmid DNA”. Kindly check the whole manuscript.
This has been changed in the whole manuscript
- Line 129-142: the % positive and total positive number for tetA, tetB, and both genes should be included in Table 2 instead of text and left the significant messages.
Percentages have been included in table 2 as reviewer requested
Discussion
- Line 153: “multidrug-resistant”
This has been corrected
- Line 154: “location”
This has been corrected
- Line 158: “multidrug resistance” or multiple antibiotic resistance”
This has been corrected
Materials and Methods
- Subsection no. must be added for each subsection title.
Subsection titles have been included as the reviewer requested
- How long the samples were taken to the laboratory and how long the samples were kept before processing? Information should be added to the manuscript.
This information has been included in the new version in the Materials and Methods subsection 4.1.
- Kindly check with the human ethic committee of your institute whether the samples taken from human sources, especially from children in this study need the ethics statement and consent form or not. It is different from the stool collection in animal sources. It is very important!!!!
- Kindly provide the data of children (age range, total number, statement from their parents to allow them as volunteer in this study, and necessary information).
Unfortunately, at the time human samples were obtained, which were kindly donated by the health care facilities in the three areas, the National Ethics Regulation was not still approved (Law number 84, May the 14th, 2019). Therefore, we did not collect the samples neither any personal information from the donors.
Round 2
Reviewer 1 Report
Thank you for including the isolation of pathogens in the methods.
Author Response
Thank you for your feedback
Reviewer 2 Report
Thank you for making the revisions. I have no further questions.
Author Response
Thank you for your feedback
Reviewer 3 Report
Comment for authors
1. The sources of human samples were not clear. As your response “at the time human samples were obtained, which were kindly donated by the health care facilities in the three areas”. I think it should be better if you say what are healthcare facilities (names and locations) provided the samples to you. In the text, you only left the message “…. and human samples came from the local people”. This statement should be clear to avoid the mistake of common research ethics.
2. Table 2 was duplicate or not? Kindly check.
3. Line 129-130; italicize tetA and tetB, and kindly check the whole manuscript.
4. The reference methods of E. coli isolation and confirmation were still unanswered. Kindly add the references for example ISO, BAM, AOAC, in-house method, etc.
Author Response
- The sources of human samples were not clear. As your response “at the time human samples were obtained, which were kindly donated by the health care facilities in the three areas”. I think it should be better if you say what are healthcare facilities (names and locations) provided the samples to you. In the text, you only left the message “…. and human samples came from the local people”. This statement should be clear to avoid the mistake of common research ethics.
The information regarding the health care facilities in each location has been included in the Materials and Methods section, as the reviewer requested.
The reviewer is right. The sentence "…. and human samples came from the local people” is misleading and has been removed in this version.
- Table 2 was duplicate or not? Kindly check.
Reviewer is right. We have delete one of the duplicates.
- Line 129-130; italicize tetAand tetB, and kindly check the whole manuscript.
This has been corrected in this version.
- The reference methods of E. coliisolation and confirmation were still unanswered. Kindly add the references for example ISO, BAM, AOAC, in-house method, etc.
The method used for the isolation and confirmation was adapted from the book: Microbiological examination of water and wastewater. This reference has been included in this version of the manuscript.
